

# Multi-directional unitarity and maximal entanglement in spatially symmetric quantum states

**Márton Mestyán[1], Balázs Pozsgay[1⋆] and Ian M. Wanless[2]**

**1** MTA-ELTE "Momentum" Integrable Quantum Dynamics Research Group,
Department of Theoretical Physics, Eötvös Loránd University, Hungary
**2** School of Mathematics, Monash University, Australia

⋆ pozsgay.balazs@gmail.com

## Abstract

We consider dual unitary operators and their multi-leg generalizations that have appeared at various places in the literature. These objects can be related to multi-party quantum states with special entanglement patterns: the sites are arranged in a spatially symmetric pattern and the states have maximal entanglement for all bipartitions that follow from the reflection symmetries of the given geometry. We consider those cases where the state itself is invariant with respect to the geometrical symmetry group. The simplest examples are those dual unitary operators which are also self dual and reflection invariant, but we also consider the generalizations in the hexagonal, cubic, and octahedral geometries. We provide a number of constructions and concrete examples for these objects for various local dimensions. All of our examples can be used to build quantum cellular automata in 1+1 or 2+1 dimensions, with multiple equivalent choices for the "direction of time".



# 1 Introduction

In this work we treat certain quantum mechanical objects with very special entanglement properties. We call these objects "multi-directional unitary operators", or alternatively "multi-directional maximally entangled states". These objects have relevance in both quantum many body physics and quantum information theory, and they have appeared in multiple works in the literature of both fields. On one hand, the objects that we treat are straightforward generalizations of the well-studied "dual unitary" (or bi-unitary) operators which appeared in the study of solvable many body systems, in particular solvable quantum cellular automata [1–3]. On the other hand, these objects are also generalizations of the "absolutely maximally entangled states" studied in quantum information theory [4]. Beyond the above two fields, the related concept of bi-unitarity also appeared much earlier in the study of von Neumann algebras [5]. In this introduction we first present the different approaches to the objects of our study, and afterwards we specify the goals of this work.

    In the field of solvable quantum many body systems, our study generalizes the concept of "dual unitarity". Dual unitary quantum circuits are solvable one dimensional many body systems which exist in discrete space and discrete time. They are local (Floquet) quantum circuits made up from the repetitions of the same two-site quantum gate, which is unitary not

only in the time, but *also in the space direction.* Such circuits describe a unitary evolution in both the time and the space dimensions. This concept of dual unitarity was formulated in [3], and it is based on earlier work on a specific model, the kicked Ising model (see [1, 2] and also [6, 7]). Besides the possibility of computing exact correlation functions [3] (including non-equilibrium situations [8]) the dual unitary models give access also to the entanglement evolution for states and also for local operators; generally, these models show maximal entanglement production for spatial bipartitions, see [2, 9–12]. Special cases of dual unitary operators are the so-called perfect tensors (with four legs), which lead to quantum circuits with maximal mixing (see [13, 14]).

Dual unitary matrices appeared much earlier in pure mathematics, namely in the study of planar algebras [5]. In this context they were called bi-unitary matrices, see for example [15]. Implications for quantum information theory were studied in [16, 17].

Generalizations of dual unitary gates to other geometries also appeared in the literature. A generalization to the triangular lattice was proposed in [18]. The extension to higher dimensional euclidean lattices with standard (hyper)-cubic geometry was mentioned already in the seminal work [3], and the case of the cubic lattice was worked out recently in [19]. The application of dual unitary gates in geometries with randomly scattered straight lines was considered in [20].

The concept of "absolutely maximally entangled state" (AME) in quantum information theory is closely related to the concept of dual unitarity and its generalizations. An AME is a multi-party state which has maximal bipartite entanglement for all possible bipartitions of the system [21–25]. An AME is sometimes also called a "perfect tensor". AMEs can be considered as a quantum mechanical extension of orthogonal arrays known from combinatorial design theory [26]. Any orthogonal array with the appropriate parameters can be used to construct a corresponding AME [24, 27, 28], but the converse is not true. There are situations when the classical object (an orthogonal array of a given type) does not exist, whereas the AME of the same type exists. A famous example is the problem of Euler's 36 officers [29, 30]; for other similar cases see [31]. An online Table of AMEs can be found at [32] (see also [33]).

AMEs can be used as quantum error correcting codes, and they appeared as ingredients in the tensor network models of the AdS/CFT correspondence: they were used as holographic error correcting codes, see [34] and the recent review [35].

Using an operator-state correspondence, an AME state of four parties (i.e., a perfect tensor with four legs) can be interpreted as a special example of a dual unitary operator. Similarly, AMEs with six or eight parties are special examples of the generalizations of dual unitary operators considered in [18] and [19]. However, the requirements of an AME are much stronger than what is needed for many applications. This led to the introduction of objects with weaker constraints: states that have maximal entanglement for a limited set of bipartitions, such that the bipartitions are selected based on some geometrical principles. In quantum information theory such objects appeared independently in [36, 37], and they were also introduced in the context of holographic error correcting codes in [38]. These generalizations involve planar arrangements of the parties of the state (legs of the tensor), just like in the case of the tri-unitary gates of [18]. On the other hand, the gates of the work [19] correspond to a three dimensional arrangement of the constituents.

In order to capture all of the above generalizations of dual unitarity and maximal entanglement, we introduce the concepts of "multi-directional unitarity" and "multi-directional maximal entanglement". Our aim is to give a general framework which encompasses all the objects mentioned above.

Among multi-directional unitary operators, our study focuses on those which are also completely invariant with respect to the symmetry group of the given geometric arrangement. In the case of dual unitarity these are the operators which are self-dual [3]; for other geome-

tries the concept has not yet been investigated. In quantum information theory similar works appeared recently, which studied quantum states with permutation invariance motivated by geometrical symmetries, see for example [39, 40]. However, these works did not investigate the states with maximal entanglement for the selected bipartitions.

The very recent work [41] considers so-called Clifford cellular automata, which are also multi-directional unitary. A common point with our work is that [41] also focuses on those cases where the arrangement possesses exact geometrical symmetries. However, there is no overlap with our work, because they only consider qubit systems with dual unitary Clifford gates, and they focus on the physical properties of the circuits. In contrast, we focus on the constructions of the gates, or equivalently, the corresponding multi-leg tensors.

In Section 2 we give all the detailed definitions, and specify the goals of this work. Afterwards, Sections 3-7 provide a number of constructions for the objects of our study. We summarize our results in Section 8, where we also present a short list of open problems. Some additional computations are included in the Appendix.

## 2 Multi-directional unitarity

In this work we consider quantum states of product Hilbert spaces, and unitary operators acting on them. In all of our cases we are dealing with homogeneous tensor products, which means that the actual Hilbert space is a tensor product of finitely many copies of $\mathbb{C}^N$ with some $N \geqslant 2$. We will work in a concrete basis. Basis elements of $\mathbb{C}^N$ are denoted as $|a\rangle$ with $a = 1, \ldots, N$, and the basis elements of $\mathbb{C}^N \otimes \mathbb{C}^N$ are denoted simply as $|ab\rangle \equiv |a\rangle \otimes |b\rangle$. Extension to tensor products with more factors follows in a straightforward way.

First we discuss the dual unitary operators, which are the simplest and most studied multi-directional unitary operators. Extensions to other geometries are given afterwards.

### 2.1 Dual unitarity

In order to introduce the dual unitary operators, we first consider unitary operators $\check{U}$ acting on the double product $\mathbb{C}^N \otimes \mathbb{C}^N$. The notation $\check{U}$ is borrowed from the theory of integrable models; we use $\check{U}$ here so that later we can also introduce another operator $U$ with a slightly different geometrical interpretation.

The matrix elements of $\check{U}$ (or other two-site operators) will be denoted as $\check{U}^{cd}_{ab}$. We use the convention

$$\check{U}|ab\rangle = \sum_{c,d=1}^{N} \check{U}^{cd}_{ab}|cd\rangle. \tag{1}$$

The operator $\check{U}$ is unitary if

$$\check{U}\check{U}^\dagger = \check{U}^\dagger\check{U} = 1, \tag{2}$$

which means in the concrete basis that

$$\sum_{a,b=1}^{N} \check{U}^{cd}_{ab}(\check{U}^{ef}_{ab})^* = \sum_{a,b=1}^{N} \check{U}^{ab}_{cd}(\check{U}^{ab}_{ef})^* = \delta^{ce}\delta^{df}. \tag{3}$$

Here the asterisk means elementwise complex conjugation. The two equalities in (2) or (3) are not independent, we write both of them for the sake of completeness.

Fig. 1 shows a pictorial representation of a dual unitary gate. Here it is understood that the "incoming" and "outgoing" spaces (with indices $a, b$ and $c, d$) are drawn as the lower and upper two legs of the gate, respectively. In this sense the picture represents time evolution in the vertical direction, upwards.

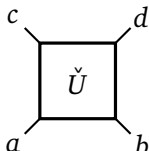

Figure 1: Pictorial representation of a dual unitary gate. In the standard interpretation $a$ and $b$ are the indices for the "incoming", and $c$ and $d$ for the "outgoing" spaces. Therefore, $\check{U}$ is seen as acting in the vertical direction upwards. In contrast, the reshuffled matrix $\check{U}^R$ acts from the left to the right, therefore its "incoming" indices are $c$ and $a$, and the outgoing indices are $d$ and $b$.

We introduce two operations for two-site gates, which amount to a "reshuffling" of indices in a chosen basis. For any two-site operator $A_{ab}^{cd}$ we define the reshuffled matrix $A^R$ and the partial transpose matrix $A^{t_1}$ via

$$(A^R)_{ca}^{db} = A_{ab}^{cd}, \qquad (A^{t_1})_{cb}^{ad} = A_{ab}^{cd}. \tag{4}$$

The operator $\check{U}$ is dual unitary if it describes unitary evolution also in the space direction. This means that the corresponding reshuffled matrix $\check{U}^R$ is also unitary. This second unitarity condition can be formulated directly for the matrix elements of the original operator $\check{U}$, giving the relations (once again dependent on each other)

$$\sum_{a,c=1}^{N} \check{U}_{ab}^{cd}(\check{U}_{ae}^{cf})^* = \sum_{a,c=1}^{N} \check{U}_{ba}^{dc}(\check{U}_{ea}^{fc})^* = \delta^{be}\delta^{df}. \tag{5}$$

The constraints (3)-(5) define an algebraic variety. An explicit parametrization of this algebraic variety is not known for $N \geq 3$. On the other hand, a complete description is available for $N = 2$ [3], see Section 3. For higher dimensions only isolated constructions are known; an extensive review of known solutions was given in [42]. A numerical method to find new solutions was presented in [43], for an early algebraic investigation see [44].

## 2.2 Dual unitarity and highly entangled states

In order to make the connection between dual unitary operators and highly entangled states, it is useful to treat the dual unitary matrices as vectors in an enlarged Hilbert space. To achieve this, we define an operator-state correspondence working in our concrete basis.

Let us consider a four-fold tensor product space

$$\mathcal{H} = V_1 \otimes V_2 \otimes V_3 \otimes V_4, \tag{6}$$

where $V_j \approx \mathbb{C}^N$ for $j = 1, \dots, 4$. We attach a geometrical interpretation to this space: we take a square and associate each space $V_j$ with a vertex of the square. The ordering is such that the numbers $1, 2, 3, 4$ are put on the square in an anti-clockwise manner, see Figure 2.

Now we map every unitary operator $\check{U}$ to a state $|\psi\rangle \in \mathcal{H}$. In our concrete basis we choose the correspondence

$$\psi_{abcd} = \frac{1}{N} \check{U}_{ab}^{dc}. \tag{7}$$

The normalization is chosen such that $\langle\psi|\psi\rangle = 1$.

It is well known that the different unitarity conditions imply that $|\psi\rangle$ has maximal entanglement for certain bipartitions. For completeness we discuss this connection in detail.

We denote the set of sites of the chain as $\mathcal{S} = \{1, 2, 3, 4\}$. Let us now consider the bipartition $\mathcal{S} = A \cup B$ with $A = \{1, 2\}$ and $B = \{3, 4\}$. The reduced density matrix of the subsystem $A$ is given by

$$\rho_A = \mathrm{Tr}_B(\rho) = \mathrm{Tr}_B(|\psi\rangle\langle\psi|) = \mathrm{Tr}_{3,4}(|\psi\rangle\langle\psi|). \tag{8}$$

By the definition of the partial trace the components are

$$(\rho_A)_{ab}^{ef} = \sum_{c,d=1}^{N} \psi_{efcd}^* \psi_{abcd}. \tag{9}$$

Combining this with (7) and the unitarity condition (3) we see that

$$(\rho_A)_{ab}^{ef} = \frac{1}{N^2} \delta_{ae} \delta_{bf}. \tag{10}$$

If the reduced density matrix is proportional to the identity then we speak of "maximal entanglement" or a "maximally mixed state". Indeed, in this case all entanglement measures obtain their maximal values. For example, the von Neumann entropy becomes

$$S_{vN} = -\mathrm{Tr}_A[\rho_A \log(\rho_A)] = \log(N^2). \tag{11}$$

Thus a unitary operator corresponds to a state having maximal entanglement between the pairs of spins (or tensor legs) $\{1, 2\}$ and $\{3, 4\}$.

The same computation can be performed also for the bipartition $\mathcal{S} = C \cup D$ with $C = \{1, 4\}$ and $D = \{2, 3\}$. In this case the components of the reduced density matrix for subsystem $C$ become

$$(\rho_C)_{ad}^{ef} = \sum_{b,c=1}^{N} \psi_{ebcf}^* \psi_{abcd}. \tag{12}$$

Comparing again with (7) we see that if the dual unitary condition (5) holds, then

$$(\rho_C)_{ad}^{ef} = \frac{1}{N^2} \delta_{ae} \delta_{df}. \tag{13}$$

Therefore, we again obtain maximal entanglement for the given bipartition.

## 2.3 Absolutely maximally entangled states

Absolutely maximal entanglement is a concept from quantum information theory, which is related to the property of dual unitarity described above. In order to define it, let us consider a generic Hilbert-space of $K$ qudits:

$$\mathcal{H} = \otimes_{j=1}^{K} V_j, \qquad V_j \approx \mathbb{C}^N. \tag{14}$$

We denote the corresponding set of sites as $\mathcal{S} = \{1, 2, \ldots, K\}$, and for simplicity we assume that $K$ is even. A state $|\psi\rangle \in \mathcal{H}$ is absolutely maximally entangled (AME) if it has maximal entanglement for every bipartition of the system. More precisely, $|\psi\rangle$ is an AME if for any bipartition $A \cup B = \mathcal{S}$ with $|A| \leqslant |B|$, the reduced density matrix $\rho_A = \mathrm{Tr}_B(|\psi\rangle\langle\psi|)$ is proportional to the identity matrix. The set of such states is generally denoted as $\mathrm{AME}(K, N)$. AMEs have been discussed in the literature in various works; selected references are [21–24, 27, 28, 33], for an online table see [32]. An AME is also called a "perfect tensor" [13]. The unitary operators that are obtained from AMEs via the operator-state correspondence were called multi-unitary [27], and specializing to cases with $K = 4, 6, \ldots$ they were called 2-unitary, 3-unitary, etc.

In the computations of the previous subsection we saw that a state $|\psi\rangle$ corresponding to a DU operator satisfies some part of the conditions of being an AME. In that case we have

$K = 4$ and the maximal entanglement criterion holds for the bipartitions $\{1, 2\} \cup \{3, 4\}$ and $\{1, 4\} \cup \{2, 3\}$. However, there is no condition for the bipartition $\{1, 3\} \cup \{2, 4\}$, which corresponds to "isolating" to the two diagonals of the square (see Figs. 1 and 2). Unitary operators $U$ for which maximal entanglement also holds for the diagonal bipartition were called "Bernoulli" in [14].

## 2.4 Multi-directional unitarity

Generalizations of the "square-shaped" (see Fig. 1) dual unitary operators to more complicated geometries such as hexagons or cubes have appeared in various places in the literature but an overall framework to describe all the various cases is lacking. In the following, we outline a general framework for these generalizations which we call "multi-directional unitary operators".

Once again we take $K$ copies of the Hilbert space $\mathbb{C}^N$, so the full Hilbert space is given by (14). In all our cases $K$ is an even number. We consider vectors $|\psi\rangle \in \mathcal{H}$ with special entanglement properties. The idea is that the states should have maximal entanglement for selected bipartitions, and the selection is motivated by certain geometric arrangements of the "parties" or sites.

The condition of maximal entanglement is the same as in the dual unitary case. Let us take a bipartition $S = A \cup B$ such that $|A| = |B| = K/2$. Once again we say that there is maximal entanglement between subsystems $A$ and $B$ if

$$\rho_A = \mathrm{Tr}_B(\rho) \sim \mathbb{1}_A, \qquad \text{and} \qquad \rho_B = \mathrm{Tr}_A(\rho) \sim \mathbb{1}_B. \tag{15}$$

The two conditions imply each other. If (15) holds, then the vector $|\Psi\rangle$ can be interpreted as a unitary operator $\mathcal{H}_A \to \mathcal{H}_B$, where $\mathcal{H}_A$ and $\mathcal{H}_B$ are the two Hilbert spaces corresponding to the two subsystems $A$ and $B$. The precise form of this operator-state correspondence is given below in Section 2.5.

The generalization of dual unitarity from the square to other geometries is the following: We arrange the sites $1, \ldots, K$ into a symmetric geometric pattern in some Euclidean space. In most of our examples we are dealing with planar arrangements but we also treat some three-dimensional arrangements. Based on the chosen arrangement, we select a list of non-empty subsets $\{A_1, A_2, A_3, \ldots, A_n\}$, $A_j \subset S$, and require that the state $|\psi\rangle$ is maximally entangled for all bipartitions

$$S = A_j \cup (S \setminus A_j), \qquad j = 1, \ldots, n. \tag{16}$$

The selection of the subsets follows from the geometric symmetries of the spatial arrangement. In all cases the bipartition is obtained by cutting the set $S$ into two equal parts in a symmetric way.

We suggest to call the states that satisfy the above requirements "multi-directional maximally entangled states", and to call the corresponding operators "multi-directional unitary operators". For the operators, the name "multi-unitary operator" would also be appropriate, but that has been already used in the context of the AME states [4, 27].

Below we give a list of the geometric arrangements that we consider in this article. Most of these arrangements have already appeared in the literature, which provides motivation to study them.

- **Square.** This corresponds to the dual unitary operators discussed above. In this case $K = 4$, the four sites are arranged as the four vertices of a square (see Fig. 2), and the subsets $A_j$ are the pairs of vertices on the four edges of the square. Therefore the allowed bipartitions correspond to cutting the square into two halves such that both halves have a pair of neighbouring vertices.

Dual unitary gates can be used to build quantum cellular automata in 1+1 dimensions. They are the simplest and most studied examples of the multi-directional unitary operators.

- **Hexagon.** In this case $K = 6$ and the sites are arranged as the vertices of a regular hexagon (see Fig. 2). The allowed bipartitions correspond to cutting the hexagon into two equal parts, such that both parts contain three neighbouring sites. The case of the hexagon was treated in [18] and the resulting operators were called "tri-unitary operators".

  The resulting operators can be used to build quantum cellular automata in 1+1 dimensions using the geometry of the triangular lattice [18].

- **Regular polygons.** In this case the sites are arranged in a plane as the vertices of a regular polygon with $K = 2k$ sides; this generalizes the previous cases of the square and the hexagon. The allowed subsets $A_j$ consist of the consecutive neighbouring $k$-tuplets (with periodic boundary conditions over the set of sites $1, \ldots, K$). The bipartitions are performed along symmetry axes that do not include any of the vertices. States that satisfy maximal entanglement in these arrangements appeared independently in (at least) three works in the literature: as "perfect tangles" in [36], as "block perfect tensors" in [38] and as "planar maximally entangled states" in [37]. See also [45] where the corresponding classical object was called "planar orthogonal array".

- **Cube.** This is our first example where the sites are arranged in a three dimensional setting. Now we have $K = 8$ sites which are put onto the vertices of a cube (see Fig. 3). The allowed subsets $A_j$ are the sets of four vertices belonging to one of the faces of the cube. Therefore, the bipartitions are obtained by cutting the cube into two halves along symmetry planes that are parallel to the faces.

  This case was considered very recently in [19], the resulting operators were called "ternary unitary". They can be used to build quantum cellular automata in 2+1 dimensions, in the arrangement of the standard cubic lattice.

- **Octahedron.** This is also an example in 3D. Now we have $K = 6$ sites which are arranged as the six vertices of an octahedron, or alternatively, as the six faces of a cube (see Fig. 3). The bipartitions are obtained by cutting the octahedron into two equal parts, each of them containing 3 vertices. Alternatively, if the vertices of the octahedron are identified as the faces of a cube, the allowed triplets of sites are obtained by selecting three faces of the cube which join each other in a chosen vertex. Note that the hexagonal arrangement also has $K = 6$, but the set of allowed bipartitions is different in the two different geometries. Below we show that the octahedral case is actually a specialization of the hexagonal case: if proper identifications are made, then the octahedral case is seen to have one more unitarity condition on top of those of the hexagonal case.

  The octahedral arrangement can also be used to build quantum cellular automata in 2+1 dimensions. The idea is to take the standard cubic lattice, put the local spaces on the faces of the cubic lattice, and to interpret a body diagonal as the direction of time. In this case the time slices are given by kagome lattices. The work [18] used this geometry, but with gates that come from the hexagonal arrangement.

- **Tetrahedron.** We add this case for completeness, even though it does not describe a new structure. Now we have $K = 4$, and the four sites are put onto the four vertices of a tetrahedron. The allowed bipartitions are the three possible ways to cut the four sites into two pairs. Therefore the resulting objects are the AMEs of four parties.

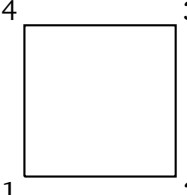 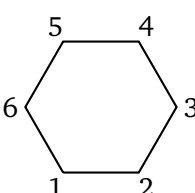

Figure 2: Labels for the local vector spaces in the square and the hexagonal geometries. The simplest multi-directional unitary operators are those which act identically along the diagonals. In the case of the square they correspond to maximally entangled two-site states prepared on the pairs of sites $(1,3)$ and $(2,4)$. For the hexagon they correspond to the Bell pairs on the pairs of sites $(1,4)$, $(2,5)$ and $(3,6)$.

In the list above we did not include the remaining two Platonic solids, the dodecahedron and the icosahedron. This is because we do not know of any application of states/operators with these geometries. However, the extension of the definition to these geometries is straightforward.

## 2.5 Identity operators and operator-state correspondence

The definition of the partial trace (15) suggests the idea to use identity operators as multi-directional unitary operators. In such cases the reduced density matrices always become identity matrices themselves (although of smaller size). However, there are various options for the operator-state correspondence, and not all of them yield multi-directional unitary operators. For example, in the case of the square geometry, $\check{U}^{cd}_{ab} = \delta^{ac}\delta^{bd}$ and $\check{U}^{cd}_{ab} = \delta^{ad}\delta^{bc}$ both seem to be legitimate identity operators. However, one can easily check that only the latter one is an actual dual unitary operator.

In the following, we use a symmetry consideration to obtain the identity operator that is multi-directional unitary. In all of the geometric arrangements considered above (except the tetrahedron), each vertex has a well defined "opposite vertex", or antipode, to which it is connected by a diagonal. The total number of diagonals that connect antipodes is $K/2$. Each such diagonal includes exactly two vertices of the arrangement, and for each allowed bipartition $A_j \cup (\mathcal{S} \setminus A_j)$ exactly one vertex is included in $A_j$ from each such diagonal. In the case of the regular polygons these diagonals are the standard diagonals that connect opposite points, whereas in the case of the cube and the octahedron they are the space diagonals. We choose the identity operator to be that which acts as identity along these diagonals. This operator is invariant under the symmetry group of the geometrical arrangement, and therefore a multi-directional unitary operator.

In the following we explicitly write down the identity operators and the corresponding states in the various geometries. By doing this, we also set the convention of the operator-state correspondence in each case. Based on this operator-state correspondence, an identity operator described above corresponds to a state made up as the product of $K/2$ maximally entangled Bell pairs prepared on the pairs of opposite vertices.

**Square geometry.** This case was already treated above, but it is useful to repeat the operator-state correspondence. Now there are $K = 4$ sites. The local Hilbert spaces will be indexed by the labels $1, 2, 3, 4$, which are associated with the four vertices of the square; we write them down in an anti-clockwise manner (see Fig. 2). Now the operator-state correspondence for $\check{U}$ is introduced via formula (7). It can be seen that $\check{U}$ acts along two edges of the square. On

the other hand, we also introduce the operator $U$ via[1]

$$\psi_{abcd} = \frac{1}{N} U_{ab}^{cd} .$$ (17)

It can be seen that $U$ acts along the diagonals of the square. The two operators are connected by the relation $\check{U} = \mathcal{P}U$, where $\mathcal{P}$ is the two-site permutation operator (SWAP gate), which acts as $\mathcal{P}(|a\rangle \otimes |b\rangle) = |b\rangle \otimes |a\rangle$.

So far the dual unitarity conditions were formulated for the elements of the matrix $\check{U}$ (5), now we also consider the conditions for the elements of $U$. It follows from the above correspondences that $U$ is dual unitary if both $U$ and $U^{t_1}$ are unitary, where $t_1$ means partial transpose with respect to the first space. As an equivalent requirement, $U$ and $U^{t_2}$ should be both unitary.

As explained above, the operator $U = 1$ is dual unitary, and it corresponds to the state

$$|\psi\rangle = \frac{1}{N} \sum_{a,b=1}^{N} |abab\rangle .$$ (18)

This is a tensor product of two Bell pairs prepared on the two diagonals of the square.

**Hexagonal geometry.** Now the $K = 6$ local spaces are associated with the 6 vertices of a hexagon; we put the labels anti-clockwise onto the vertices (see Fig. 2). The standard operator-state correspondence is through the relation

$$\psi_{abcdef} = \frac{1}{\sqrt{N^3}} \check{U}_{abc}^{fed} .$$ (19)

This corresponds to the definitions in [18]. Alternatively, we use the correspondence

$$\psi_{abcdef} = \frac{1}{\sqrt{N^3}} U_{abc}^{def} .$$ (20)

The connection between the two definitions is given by

$$U = \mathcal{P}_{1,3} \check{U} ,$$ (21)

where $\mathcal{P}_{1,3}$ permutes spaces 1 and 3. In these conventions an operator $U$ is multi-directional unitary if $U$, $U^{t_1}$ and also $U^{t_3}$ are unitary.

The state corresponding to the identity operator $U = 1$ is given by

$$|\psi\rangle = \frac{1}{N^{3/2}} \sum_{a,b,c=1}^{N} |abcabc\rangle .$$ (22)

This is a product of three Bell pairs prepared on the long diagonals of the hexagon.

---

[1]The reader might find it confusing that we introduced two different unitary operators for the same object; the reasons for defining both $U$ and $\check{U}$ are as follows. The dual unitary quantum circuits always use the two-site unitary operator $\check{U}$. On the other hand, the mathematical properties are more simple when we formulate them for the operator $U$. For example, it is shown in the main text that $U$ can be chosen as the identity operator.

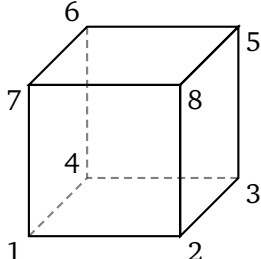 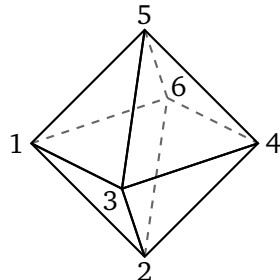

Figure 3: Labels for the vector spaces in the cubic and octahedral geometries. The simplest multi-directional unitary operators are those which act as the identity matrix along the body diagonals. In terms of the quantum states this corresponds to preparing maximally entangled two-site states on the pairs of sites $(1, 5)$, $(2, 6)$, $(3, 7)$ and $(4, 8)$ in the cubic, and $(1, 4)$, $(2, 5)$ and $(3, 6)$ in the octahedral geometries.

**Cubic geometry.** Now we have $K = 8$ sites, and we introduce the following labelling for them: the sites $1, 2, 3, 4$ are placed on a selected face in the given order, and the sites $5, 6, 7, 8$ are on the other face, such that the sites $j$ and $j + 4$ with $j = 1, \ldots, 4$ are on the same body diagonal (see Figure 3). Again there are two possibilities for an operator-state correspondence. We introduce the matrix $\check{U}$ which acts from the tensor product of the spaces $1, 2, 3, 4$ to the product of spaces $5, 6, 7, 8$, such that the "incoming" and "outgoing" spaces are connected by the edges of the cube. This gives the operator-state correspondence

$$\psi_{a_1 a_2 a_3 a_4 a_5 a_6 a_7 a_8} = \frac{1}{N^2} \check{U}^{a_7 a_8 a_5 a_6}_{a_1 a_2 a_3 a_4}, \qquad a_j = 1, \ldots, N, \qquad j = 1, \ldots, 8. \tag{23}$$

The $\check{U}$ operator is used to build the quantum circuits in 2+1 dimensional space, see [19]. On the other hand, we again introduce an operator $U$ which acts along the body diagonals. In this case the operator-state correspondence is

$$\psi_{a_1 a_2 a_3 a_4 a_5 a_6 a_7 a_8} = \frac{1}{N^2} U^{a_5 a_6 a_7 a_8}_{a_1 a_2 a_3 a_4}, \qquad a_j = 1, \ldots, N, \qquad j = 1, \ldots, 8. \tag{24}$$

In these conventions an operator $U$ is multi-directional unitary if $U$, $U^{t_2 t_3}$ and $U^{t_3 t_4}$ are all unitary.

The identity operator can be constructed in the same way as for the square and the hexagonal geometries. The corresponding state will be a product of four Bell pairs over the space diagonals of the cube.

**Octahedral geometry.** Now we have $K = 6$ sites, and we label them according to the pattern shown in Fig. 3. Now the pairs $(1, 4)$, $(2, 5)$ and $(3, 6)$ are the body diagonals. In accordance, we introduce the operator-state correspondence as

$$\psi_{abcdef} = \frac{1}{\sqrt{N^3}} U^{def}_{abc}. \tag{25}$$

In these conventions an operator $U$ is multi-directional unitary if $U$, $U^{t_1}$, $U^{t_2}$ and $U^{t_3}$ are all unitary.

Let us compare these conditions with those of the hexagonal case where we also have $K = 6$. We see that the octahedral conditions are stronger: they include one extra condition on top of those of the hexagonal case. This extra condition is the unitarity of $U^{t_2}$. It follows that every octahedral unitary matrix is hexagonal unitary, but the converse is not true. The

similarity between these constraints was used in [18], where a 2+1 dimensional quantum circuit was built using the hexagonal unitary gates.

In the octahedral case, the identity operator can be constructed in the same way as for the square and the hexagonal geometries. The corresponding state will be a product of three Bell pairs over the diagonals of the octahedron.

## 2.6 Symmetries and equivalence classes

Let us discuss the operations on the states $|\psi\rangle$ which preserve the required entanglement properties. These operations will be used to obtain classifications of the various multi-directional unitary operators.

Among the operations, we distinguish two classes. The first class is the class of local unitary operations under which all entanglement measures are invariant. Consequently, if $|\psi\rangle$ satisfies the constraints of multi-directional unitarity in one of the geometries, then the state

$$(U^{(1)} \otimes U^{(2)} \otimes \cdots \otimes U^{(K)})|\psi\rangle\,, \qquad U^{(j)} \in SU(N)\,, \quad j = 1, 2, \ldots, K\,, \tag{26}$$

also satisfies them. If two vectors $|\psi_1\rangle$ and $|\psi_2\rangle$ can be transformed into each other in this way then we say that they are local unitary (LU) equivalent.

The second class of operations consists of those spatial rearrangements (permutations) of the sites which respect the geometry at hand. We denote the corresponding symmetry group as $G$. In the simplest case of $K = 4$ and the geometry of the square the symmetry transformations are generated by rotations of degree $\pi/2$ and a reflection, and the symmetry group is the dihedral group $G = D_4$. In the case of regular polygons the spatial symmetries are given by the dihedral group $G = D_K$, whereas for the cube and the octahedron the spatial symmetries are given by the octahedral group $G = O_h \sim S_4 \otimes S_2$. Every symmetry transformation $g \in G$ can be understood as a permutation of the sites, which leads to a permutation operation $\mathcal{P}_g$ on the tensor product of the individual vector spaces. It is clear that if $|\psi\rangle$ satisfies the entanglement criteria, then $\mathcal{P}_g|\psi\rangle$ also satisfies them for every $g \in G$.

We say that two states $|\psi_1\rangle$ and $|\psi_2\rangle$ are equivalent if they can be transformed into each other using a combination of a permutation $\mathcal{P}_g$ and local unitary operations.

## 2.7 Spatially symmetric states

In this work we consider those multi-directional maximally entangled states (or multi-directional unitary operators) that are also maximally symmetric with respect to the corresponding spatial symmetry group. This means that for any $g \in G$ the states satisfy

$$\mathcal{P}_g|\psi\rangle = |\psi\rangle\,. \tag{27}$$

In this work we call these states "spatially symmetric". The spatially symmetric states have the appealing property that their concrete representation does not depend on a chosen orientation of the geometric arrangement. Furthermore, if we require the symmetry (27) then the property of maximal entanglement has to be checked only in one of the allowed bipartitions. In practice this means that one needs to check the unitarity of only one matrix, because the matrix of the operator remains the same under all the spatial symmetry transformations.

Once the permutation symmetry (27) is required, the general local unitary transformations (26) are not allowed anymore. However, the symmetric unitary transformations

$$|\psi\rangle \quad \rightarrow \quad (V \otimes V \otimes \cdots \otimes V)|\psi\rangle\,, \qquad V \in SU(N)\,, \tag{28}$$

are still allowed. We say that two spatially symmetric states $|\psi_1\rangle$ and $|\psi_2\rangle$ are equivalent if there is a one-site operator $V$ which transforms them into each other.

Our requirement (27) for the spatial symmetry is rather strong, because it requires invariance in a strict sense. We could also require the weaker condition that for each $g \in G$ the state $\mathcal{P}_g|\psi\rangle$ should be LU equivalent to $|\psi\rangle$. For completeness we show two examples for this in Appendix A.

In the case of dual unitary matrices certain solutions with other types of symmetry properties have already been considered in the literature. Perfect tensors with $SU(2)$ invariance were studied in [46], whereas dual unitary matrices with diagonal $SU(N)$ or diagonal $SO(N)$ symmetries were analyzed in [47–49]. However, the spatial symmetry that we consider has not yet been investigated in the context of these maximally entangled states.

## 2.8 Further entanglement properties

The definition of multi-directional unitarity enforces maximal entanglement along selected bipartitions. States satisfying these requirements can still have very different bipartite or multipartite entanglement patterns.

As it was explained above, the identity operations along the diagonals are multi-directional unitary. They correspond to states with maximally entangled pairs prepared on the diagonals. In these states the diagonals are completely independent from each other: there is no entanglement between them. To be more precise: if we group the two sites on each diagonal into a macro-site of dimension $N^2$, then we obtain a product state. In the quantum circuits that are built out of these objects [3,18,19] the identity matrices describe free information propagation in selected directions of the circuit. Interaction happens in the circuit only if the diagonals of the state are entangled with each other.

Therefore it is very natural to consider the bipartite or multi-partite entanglement entropies *between the diagonals*. Non-zero entanglement between the diagonals signals that the state $|\psi\rangle$ is non-trivial, and the corresponding operator $\check{U}$ generates interactions in the quantum circuits. In this work we focus on these non-trivial cases.

# 3 Dual unitary matrices

In the previous Section, we have laid the general framework for treating multi-directional unitary operators. We have also restricted our attention to operators that are invariant under the spatial symmetries of the arrangement. In the rest of the article, we discuss their various special cases.

First we consider the square geometry and dual unitary matrices. Invariance with respect to the geometrical symmetry operations means that the matrix of $U$ coincides with the 7 reshuffled matrices that arise from the 7 non-identical transformations of the square. The symmetry group is generated by a rotation and a reflection, therefore it is enough to check invariance with respect to those symmetries.

For the matrix $\check{U}$ these two transformations lead to

$$\check{U} \to \check{U}^R, \qquad \check{U} \to \mathcal{P}\check{U}\mathcal{P}, \tag{29}$$

where the definition of $\check{U}^R$ is given by (4). Therefore, a symmetric matrix is one which satisfies

$$\check{U} = \check{U}^R = \mathcal{P}\check{U}\mathcal{P}. \tag{30}$$

In terms of the matrix $U$ the two necessary relations can be chosen as

$$U = U^{t_1} = \mathcal{P}U\mathcal{P}. \tag{31}$$

Matrices that satisfy the properties (30) or (31) are called self-dual.

Note that our conventions imply that the time reflection step does not involve any complex conjugation, and it is given merely by the full transpose of the operators. This is to be contrasted with the usual choice in quantum mechanics, where time reflection also involves a complex conjugation. However, our choice is natural here, because we are treating a space-time duality, and time reflection in a given direction describes space reflection in the other direction, therefore it is most natural that it does not involve a complex conjugation.

The symmetric unitary transformations (28) result in

$$U \quad \rightarrow \quad (V^t \otimes V^t) U (V \otimes V). \tag{32}$$

The transposition in the factors on the left follows simply from the operator-state correspondence, and the symmetric prescription (28) for the transformation of the states.

## 3.1 Two qubits

Let us consider the special case of $N = 2$, i.e., operators acting on two qubits. In this case, dual unitary gates were completely classified in [3] based on the Cartan decomposition of a general $U(4)$ matrix. It was found in [3] that a generic dual unitary matrix can be written as

$$U = e^{i\varphi}(S_1 \otimes S_2) D (S_3 \otimes S_4), \tag{33}$$

where $S_{1,2,3,4} \in SU(2)$ are one-site operators, $\varphi \in \mathbb{R}$, and $D$ is a diagonal unitary matrix, for which a parametrization can be chosen as

$$D = e^{i\alpha Z_1 Z_2} = \cos(\alpha) + i\sin(\alpha) Z_1 Z_2 = \mathrm{diag}(e^{i\alpha}, e^{-i\alpha}, e^{-i\alpha}, e^{i\alpha}), \qquad \alpha \in \mathbb{R}, \tag{34}$$

where $Z_j$ stands for the Pauli matrix $\sigma^z$ acting on site $j$. Note that we gave the representation of the matrix $U$, and not $\breve{U}$.

Now we consider the self-dual cases. First of all we observe that the matrix $D$ is self-dual for every $\alpha$: it is easy to see that it satisfies the two conditions of (31). From the form (32) of the self-duality preserving transformations it follows that self-dual solutions are produced as

$$e^{i\varphi}(V^t \otimes V^t) D (V \otimes V), \qquad V \in SU(2). \tag{35}$$

This gives a large family of self-dual solutions, with a total number of 5 real parameters: 3 parameters of $V$, the "inner phase" $\alpha$ and the "global phase" $\varphi$.

It can be shown that (35) covers all self-dual cases. Every dual unitary matrix with $N = 2$ is of the form (33), and the decomposition is unique if $\alpha \neq 0$. The spatial symmetry operations simply just exchange the external one-site operators, therefore the invariance property implies the form (35). In the special case $\alpha = 0$ we are dealing with a simpler situation, namely $U = (S_1 S_3) \otimes (S_2 S_4)$. Such a decomposition is clearly not unique, but in the case of $U$ being self-dual, the form (35) can be established nevertheless.

A very important special case is the evolution matrix of the *kicked Ising model* at the self-dual point [1, 2, 6, 7, 50, 51]. It can be written as

$$\breve{U} = e^{i\frac{\pi}{4} Z_1 Z_2} e^{i\frac{\pi}{4}(X_1 + X_2)} e^{i\frac{\pi}{4} Z_1 Z_2}. \tag{36}$$

Note that now we gave the operator $\breve{U}$. In a concrete matrix representation we have

$$\breve{U} = \frac{i}{2} \begin{pmatrix} 1 & -1 & -1 & -1 \\ -1 & -1 & 1 & -1 \\ -1 & 1 & -1 & -1 \\ -1 & -1 & -1 & 1 \end{pmatrix}. \tag{37}$$

This is proportional to a real Hadamard matrix. We will return to this specific matrix later, because it is the simplest example for two general constructions that we treat in Sections 5 and 6.

# 4 Diagonal unitary matrices

The simplest idea for multi-directional unitary operators is to take the identity matrices (with the proper operator-state correspondence laid out above) and to dress them with phase factors. For the dual unitary matrices this idea was explored in detail in [52], whereas in the hexagonal geometry it appeared in [18]. Here we also discuss this construction, by focusing on the spatially symmetric solutions. Below we give the details in the square and hexagonal geometries; the extensions to the cube and the octahedron are straightforward.

## 4.1 Dual unitary matrices

In the case of dual unitary matrices (square geometry) we take $U = D$ with $D$ being a diagonal unitary matrix with the non-zero elements being

$$D_{ab}^{ab} = e^{i\varphi_{ab}}, \qquad \varphi_{ab} \in \mathbb{R}, \quad a, b = 1, \dots, N. \tag{38}$$

As first explained in [52], such a matrix is dual unitary for arbitrary phases and arbitrary $N$. This follows immediately from the fact that the partial transpose of a diagonal matrix is itself.

An alternative "physical interpretation" can be given as follows: the gate $D$ describes a scattering event of two particles on two crossing world lines. The two particles have an inner degree of freedom, which is indexed by the labels $a, b = 1, \dots, N$. The incoming and outgoing particles have the same label, and there is a phase factor associated with each event. This phase depends on the pairs of labels $(a, b)$.

In order to obtain a spatially symmetric (self-dual) solution one needs to check the symmetry condition, which is the second relation in (31). In this concrete case one obtains the simple symmetry relation

$$\varphi_{ab} = \varphi_{ba}. \tag{39}$$

Thus we obtain a large family of self-dual solutions, with the number of real parameters being $N(N + 1)/2$. However, we can set the phases $\varphi_{aa}$ to zero with symmetrically placed one-site operators, therefore the total number of intrinsically independent phases is $N(N - 1)/2$.

For completeness we also give the four site state corresponding to (38). It reads

$$|\psi\rangle = \frac{1}{N} \sum_{a,b=1}^{N} e^{i\varphi_{ab}} |a, b, a, b\rangle. \tag{40}$$

If $\varphi_{aa} = 0$ but $\varphi_{ab} \neq 0$ for some $a, b$, then these states have entanglement of the diagonals, therefore they generate interactions in the quantum circuits. The effects of these interactions were explored in detail in [52].

## 4.2 Hexagonal geometry

We use the notations introduced in Section 2.5 for the case of the hexagonal geometry. In this case we treat a diagonal matrix $U = D$, where now the matrix elements are

$$D_{abc}^{abc} = e^{i\varphi_{abc}}, \qquad \varphi_{abc} \in \mathbb{R}, \quad a, b, c = 1, \dots, N. \tag{41}$$

It is easy to see that a full hexagonal symmetry is achieved only if $\varphi_{abc}$ is completely symmetric with respect to $a, b, c$. Generally this gives a large family of solutions. For completeness we also give the six site state corresponding to (41). It reads

$$|\psi\rangle = \frac{1}{N^{3/2}} \sum_{a,b,c=1}^{N} e^{i\varphi_{abc}} |abcabc\rangle. \tag{42}$$

In the case of qubits ($N = 2$) the solutions can be parametrized as

$$D = \exp\left[i(\alpha + \beta(Z_1 + Z_2 + Z_3) + \gamma(Z_1 Z_2 + Z_2 Z_3 + Z_1 Z_3) + \delta Z_1 Z_2 Z_3)\right]. \tag{43}$$

The parameters $\alpha$ and $\beta$ control a global phase and three one-site unitary operations, and we are free to set them to 0. The remaining "inner" parameters are $\gamma$ and $\delta$. If $\delta = 0$ then the three site gate factorizes as the product of two-site operators

$$D_{123} = E_{12} E_{13} E_{23}, \qquad E_{jk} = e^{i\gamma Z_j Z_k}. \tag{44}$$

However, if $\delta \neq 0$ then a true three-body interaction term appears, which couples the three diagonals of the hexagon. For $\gamma \neq 0$ and/or $\delta \neq 0$ there is non-zero entanglement between the diagonals of the hexagon.

## 5  Constructions from Hadamard matrices

It is known that dual unitary matrices can be constructed from complex Hadamard matrices. The essence of this method appeared in many places in the literature [1, 6, 14, 31, 51], but its most general formulation was given in [53]. Here we explore this idea once more, extending it also to the cubic geometry.

A complex Hadamard matrix $H$ of size $N \times N$ has the following two properties: all its matrix elements are complex numbers of modulus one, and the matrix is proportional to a unitary matrix. The review [54] discusses the history of research on Hadamard matrices, their applications to combinatorial design theory and quantum information theory, and it also gives a number of constructions and concrete examples for small sizes.

Let us also discuss the simplest examples for Hadamard matrices. First we note that the Hadamard matrices form equivalence classes: two Hadamard matrices $A$ and $A'$ are in the same class if there are diagonal unitary matrices $D_1$ and $D_2$ and permutation matrices $P_1$ and $P_2$ such that

$$A' = P_2 D_2 A D_1 P_1. \tag{45}$$

Generally it is enough to discuss examples up to such equivalence steps.

For $N = 2$ it is known that the only equivalence class is given by the real Hadamard matrix

$$A = \begin{pmatrix} 1 & 1 \\ 1 & -1 \end{pmatrix}. \tag{46}$$

For $N = 3$ there is again only one equivalence class, which is given by

$$A = \begin{pmatrix} 1 & 1 & 1 \\ 1 & \omega & \omega^2 \\ 1 & \omega^2 & \omega \end{pmatrix}, \qquad \omega = e^{2i\pi/3}. \tag{47}$$

For a generic $N$ a symmetric complex Hadamard matrix is the Fourier matrix, given by

$$F_{jk} = e^{\frac{2i\pi}{N}(j-1)(k-1)}, \qquad j, k = 1, \ldots, N. \tag{48}$$

The matrices (46) and (47) are special cases of the Fourier matrix at $N = 2$ and $N = 3$.

## 5.1 Dual unitary matrices

In the case of square geometry, let us take four Hadamard matrices $A, B, C, E$ of size $N \times N$ and construct a new matrix $H$ of size $N^2 \times N^2$ (acting on the two-fold tensor product space) through the formula

$$H_{ab}^{dc} = A_a^b B_b^c C_c^d E_d^a . \tag{49}$$

There is no summation over $a, b, c, d$; this is just a product of the given matrix elements. Clearly we get $|H_{ab}^{cd}| = 1$. The geometrical interpretation is the following: if we regard the sites as the four vertices of a square, then the small Hadamard matrices are attached to the four edges of the same square, and multiplication is taken component-wise.

It is easy to show that the matrix $\check{U} = H/N$ is unitary. To see this, we rewrite the matrix $\check{U}$ as

$$\check{U} = \frac{1}{N} D^C (E^t \otimes B) D^A , \tag{50}$$

where $E$ and $B$ are the operators as given by the original Hadamard matrices, $^t$ denotes transpose, and $D^A$ and $D^C$ are diagonal two-site operators given by

$$(D^A)_{ab}^{ab} = A_a^b , \qquad (D^C)_{dc}^{dc} = C_c^d . \tag{51}$$

It follows from the Hadamard properties that $D^A$ and $D^C$ are unitary, and supplemented with the unitarity of $B/\sqrt{N}$ and $E^t/\sqrt{N}$ we see that $\check{U}$ is indeed unitary. Essentially the same proof was given in [53].

The construction (49) is invariant with respect to the symmetries of the square: the transformations lead to transpositions and permutations of the factors, but they do not change the structure of the formula. Therefore, these $\check{U}$ matrices are also dual unitary.

Solutions which are invariant with respect to the geometrical symmetries are found when all matrices are equal and symmetric. Thus we choose a single complex Hadamard matrix $A$ satisfying $A = A^t$ and we write

$$\check{U}_{ab}^{dc} = \frac{1}{N} A_a^b A_b^c A_c^d A_d^a . \tag{52}$$

Each such $\check{U}$ is dual unitary and self dual.

In the case of $N = 2$ we consider the Hadamard matrix

$$A' = \begin{pmatrix} 1 & i \\ i & 1 \end{pmatrix} . \tag{53}$$

This is in the same equivalence class as $A$ in (46). It can be checked by direct computation that the gate of the kicked Ising model (37) is obtained from this matrix via (52) (up to an overall phase).

For generic $N$ the quantum circuits arising from the Fourier matrix were considered earlier in [51].

## 5.2 Cubic geometry

The previous construction can be extended to the cubic geometry. Now we take 12 copies of a symmetric complex Hadamard matrix $A$ and "attach" them to the 12 edges of the cube. The components of the operator $\check{U}$ are then obtained by taking the products of matrix elements of $A$. An explicit formula for the spatially symmetric case is written down as

$$\check{U} = D_{14}^A D_{34}^A D_{23}^A D_{12}^A A_4 A_3 A_2 A_1 D_{14}^A D_{34}^A D_{23}^A D_{12}^A , \tag{54}$$

where $D^A$ is defined in (51). Each factor in this product represents one of the edges of the cube. The formula is a generalization of (50). Unitarity is proven in a straightforward way, and the spatial symmetry follows from the construction.

The simplest case is that of the kicked Ising model ($N = 2$) in 2+1 dimensions, which was considered in [55]. In this case the chosen Hadamard matrix is again given by (53), and the operator $\check{U}$ acting on 4 qubits can be written as

$$\check{U} = e^{i\frac{\pi}{4}(Z_1 Z_2 + Z_2 Z_3 + Z_3 Z_4 + Z_4 Z_1)} e^{i\frac{\pi}{4}(X_1 + X_2 + X_3 + X_4)} e^{i\frac{\pi}{4}(Z_1 Z_2 + Z_2 Z_3 + Z_3 Z_4 + Z_4 Z_1)}. \tag{55}$$

This operator can be written as $\check{U} = \frac{1}{2^{3/2}} H$, where $H$ is a complex Hadamard matrix of size $16 \times 16$. This matrix is equivalent to a real Hadamard matrix, which can be seen from the equivalence between (46) and (53).

## 6  Graph states

Graph states are well known entanglement resources in quantum information theory. The main idea behind them is to take a Hilbert space of $K$ sites, to start from a product state, and apply a set of commuting entangling operations which can be encoded in a graph with $K$ vertices. In this work we apply the formalism and the main results of [56]. We assume that the local dimension $N$ is a prime number; extensions of the method to prime power dimensions were treated in [57].

A graph state $|\psi\rangle$ on $K$ sites is characterized by a "colored" (or labeled) undirected graph with $K$ vertices. Each edge of the graph can take $N$ values (labels) given by $\{0, 1, 2, \ldots, N-1\}$; these values are encoded in the incidence matrix $I_{jk}$, where $j \neq k$ stand for two vertices. Self loops are disregarded.

The construction starts with a product state $|\psi_0\rangle$ which is prepared as

$$|\psi_0\rangle = \otimes_{j=1}^{K} |e\rangle_j, \qquad |e\rangle = \frac{1}{\sqrt{N}} \left( \sum_{a=1}^{N} |a\rangle \right). \tag{56}$$

We act on this product state with a sequence of two-site gates. We introduce the so-called controlled-$Z$ gate, which is a diagonal two-site operator with matrix elements

$$\mathcal{Z}_{ab}^{ab} = (\omega)^{a \cdot b}, \qquad \omega = e^{2i\pi/N}. \tag{57}$$

The matrix elements of $\mathcal{Z}$ are identical with those of the Fourier matrix (48) (up to an overall constant), but now the $N \times N$ matrix elements are arranged in a diagonal matrix of size $N^2 \times N^2$. In a certain sense the matrix $\mathcal{Z}$ is the dual of the Fourier matrix (48). The transposition symmetry of the matrix $F$ translates into a space reflection symmetry of $\mathcal{Z}$.

Let us now draw a graph on the $K$ vertices with incidence matrix $I$. Then the corresponding graph state is

$$|\psi\rangle = \prod_{1 \leqslant j < k \leqslant K} (\mathcal{Z}_{jk})^{I_{jk}} |\psi_0\rangle. \tag{58}$$

Here $\mathcal{Z}_{jk}$ stands for the controlled-$Z$ operator acting on sites $j$ and $k$. These operators are all diagonal in the given basis, therefore their product is well defined without specifying the ordering. The resulting state is such that each component has equal magnitude, and entanglement arises from the various phases that can appear.

The graph state is characterized by the incidence matrix $I$. States corresponding to two different graphs can be LU equivalent; this is treated for example in [58, 59]. It is important that the formula (58) does not depend on $N$, but the vectors $|\psi_0\rangle$ and the operator $\mathcal{Z}$ are different for different values of $N$. Therefore, the same graph can encode states with very different types of entanglement as $N$ is varied. We also note that in certain cases the graph states are LU equivalent to "classical solutions" which are treated in Section 7; for the treatment of these connections see for example [25, 31].

The entanglement properties of the states (58) can be found from certain properties of the incidence matrix. It was shown in [56] that the state is maximally entangled with respect to a bipartition $S = A \cup B$ if a certain reduced incidence matrix $I^{AB}$ has maximal rank. To be more concrete, let us take a graph with $K$ vertices, and divide the sets of vertices into two subsets $A$ and $B$ with $K/2$ sites. The reduced incidence matrix $I^{AB}$ of size $K/2 \times K/2$ is determined by the links connecting the different points in $A$ and $B$: if we order the list of sites such that the first $K/2$ sites are in $A$ and the second $K/2$ sites are in $B$ then the original incidence matrix is given in block form as

$$I = \begin{pmatrix} I^{AA} & I^{AB} \\ I^{BA} & I^{BB} \end{pmatrix}, \qquad I^{BA} = (I^{AB})^t. \tag{59}$$

Maximal rank means that the rows (or columns) of $I^{AB}$ are linearly independent vectors over $\mathbb{Z}_N$, or alternatively, that the determinant of the matrix is non-zero in $\mathbb{Z}_N$. In practice this is equivalent to the condition

$$\det I^{AB} \neq 0 \bmod N. \tag{60}$$

It is very important that the condition is understood in $\mathbb{Z}_N$: some graph states can be maximally entangled for certain values of $N$, and less entangled for other values; examples are treated in [56].

It is easy to find a condition for the spatial invariance (27): The graph must be invariant with respect to the given geometric symmetry group. This requirement dramatically decreases the number of possibilities, but it also enables a simple direct check of the maximal entanglement. Below we analyze the solutions in the different geometries.

In all our cases we choose a special convention for the reduced density matrix $I^{AB}$. We divide the set of $K$ vertices into two subsets $A = \{1, \ldots, K/2\}$ and $B = \{K/2 + 1, \ldots, K\}$, and we set the matrix element $(I^{AB})_{jk}$ with $j, k = 1, \ldots, K/2$ to be equal to the label of the edge connecting the sites with indices $j$ and $k+K/2$. We use the ordering of the site indices explained in Subsection 2.5.

## 6.1 Dual unitary matrices

Again we are dealing with the geometry of the square. Spatially invariant graphs on the square have just two free parameters: the labels for the edges and for the diagonals of the square. Denoting them by $\alpha$ and $\beta$, respectively, we get the reduced incidence matrix

$$I^{AB} = \begin{pmatrix} \beta & \alpha \\ \alpha & \beta \end{pmatrix}. \tag{61}$$

There is maximal entanglement if $\beta^2 - \alpha^2 \neq 0 \bmod N$. This leads to the two conditions

$$\beta + \alpha \neq 0 \bmod N, \quad \text{and} \quad \beta - \alpha \neq 0 \bmod N. \tag{62}$$

For $N = 2$ there are only two choices. The case of $\beta = 0$ and $\alpha = 1$ is a special case of the construction (52) with Hadamard matrices. The case of $\alpha = 0$ and $\beta = 1$ is simply just the identity along the diagonals dressed with one-site unitary operators.

For $N = 3$ the two conditions imply that either $\alpha = 0$ or $\beta = 0$, leading to essentially the same configurations as for $N = 2$.

New solutions appear for $N > 3$. For example $\alpha = 1$ and $\beta = 2$ is a solution for every prime $N > 3$.

Dual unitary matrices constructed from graph states were considered earlier in [14, 31, 42]; in [14, 31] they were called quantum cat maps. It is known that for $N = 3$ there is a graph state, which is an AME, but it does not have spatial invariance in the strict sense [56]. We treat its classical version in Appendix A, where we also show that it is spatially invariant in a weaker sense.

## 6.2 Hexagonal geometry

Now a graph state has three parameters. They correspond to the three types of vertices: those connecting nearest neighbours, next-to-nearest neighbours, and sites on diagonals. Denoting the corresponding labels (in this order) as $\alpha, \beta, \gamma$ we obtain the reduced incidence matrix

$$I^{AB} = \begin{pmatrix} \gamma & \beta & \alpha \\ \beta & \gamma & \beta \\ \alpha & \beta & \gamma \end{pmatrix}. \tag{63}$$

Now we find the determinant

$$\det I^{AB} = (\gamma - \alpha)(\gamma^2 - 2\beta^2 + \gamma\alpha). \tag{64}$$

Thus we get the two requirements

$$\alpha - \gamma \neq 0 \bmod N, \quad \text{and} \quad \gamma^2 - 2\beta^2 + \gamma\alpha \neq 0 \bmod N. \tag{65}$$

For $N = 2$ these requirements simplify to

$$\alpha - \gamma \neq 0 \bmod N, \quad \text{and} \quad \gamma(\gamma + \alpha) \neq 0 \bmod N. \tag{66}$$

This dictates $\alpha = 0$ and $\gamma = 1$, and in this case $\beta$ can be chosen to be either 0 or 1. The resulting cases are:

- The solution with $\alpha = \beta = 0$ and $\gamma = 1$ is trivial, it corresponds to the identity along the diagonals plus one-site unitary operators.

- The solution $\alpha = 0$ and $\beta = \gamma = 1$ is non-trivial. More detailed computations show that it is an AME. It was well known that an AME with 6 qubits exists, and in fact a graph isomorphic to the solution $\alpha = 0$ and $\beta = \gamma = 1$ was presented in [56], see Fig. 2.e in that work. However, the hexagonal symmetry of that given graph was not pointed out there.

For $N \geq 3$ there are more solutions, and there are some solutions which work for every prime $N$. For example, the choice $\alpha = 1$, $\beta = 1$ and $\gamma = 0$ always satisfies (65).

## 6.3 Cubic geometry

Now the graph consists of $K = 8$ sites and there are 3 types of edges of the graph: those which correspond to the edges, the face diagonals and the body diagonals of the cube. Denoting the corresponding labels as $\alpha, \beta, \gamma$ the reduced incidence matrix for a bipartition into two faces is written as

$$I^{AB} = \begin{pmatrix} \gamma & \beta & \alpha & \beta \\ \beta & \gamma & \beta & \alpha \\ \alpha & \beta & \gamma & \beta \\ \beta & \alpha & \beta & \gamma \end{pmatrix}. \tag{67}$$

Now the determinant is

$$\det I^{AB} = (\alpha - \gamma)^2(\alpha - 2\beta + \gamma)(\alpha + 2\beta + \gamma). \tag{68}$$

For $N = 2$ the determinant simplifies further to

$$\det I^{AB} = (\alpha - \gamma)^4 \bmod 2. \tag{69}$$

Therefore the value of $\beta$ is irrelevant, and we get the only condition that $\alpha$ and $\gamma$ need to be different. This leads to the following four solutions:

- $\gamma = 1$ and $\alpha = \beta = 0$. This is the trivial solution, with non-interacting body diagonals.

- $\alpha = 1$ and $\beta = \gamma = 0$. Now there are controlled $Z$-operators attached to the edges of the cube. This is the same state that is found via the Hadamard matrices in Section 5.2.

- $\alpha = 0$, $\beta = \gamma = 1$ and $\alpha = \beta = 1$, $\gamma = 0$. These are non-trivial new solutions.

Naturally, for $N \geqslant 3$ there are more solutions, which can be found easily by numerical inspection.

### 6.4 Octahedral geometry

Now there are $K = 6$ sites and only two types of edges of the graph: those corresponding to the edges and the body diagonals of the octahedron. Denoting their labels as $\alpha$ and $\gamma$, the reduced incidence matrix is

$$I^{AB} = \begin{pmatrix} \gamma & \alpha & \alpha \\ \alpha & \gamma & \alpha \\ \alpha & \alpha & \gamma \end{pmatrix}. \tag{70}$$

We have

$$\det I^{AB} = (\gamma - \alpha)^2 (\gamma + 2\alpha). \tag{71}$$

For $N = 2$ this simplifies to

$$\det I^{AB} = (\gamma - \alpha)^2 \gamma \bmod 2. \tag{72}$$

Therefore the only choice is $\gamma = 1$, $\alpha = 0$, which is the trivial solution with identity operations along the diagonals.

For $N \geqslant 3$ there are more solutions. Interestingly, the solution $\alpha = 1$, $\gamma = 0$ works for all prime $N \geqslant 3$. In this case there is a controlled-$Z$ operator for each edge of the octahedron.

We can see that the octahedral case is a specialization of the hexagonal case: the incidence matrix (70) coincides with the special case $\alpha = \beta$ of (63).

## 7 Classical solutions

We call a unitary operator "deterministic" or "classical" if it is represented by a permutation matrix in a selected basis. When the operator is applied successively, it gives rise to a deterministic walk over the elements of this basis. In this section we consider multi-directional operators that have the classical property, i.e., they act as permutations of basis elements in more than one direction. Instead of giving a general treatment right away we consider the different geometries case by case.

We begin with the square geometry ($K = 4$). Now a unitary operator is represented by a matrix, whose matrix elements we denote as $U^{b_1 b_2}_{a_1 a_2}$. The deterministic property combined with unitarity means that for all pairs of numbers $(a_1, a_2)$ with $a_1, a_2 \in \{1, \ldots, N\}$ there is a unique pair $(b_1, b_2)$ with $b_1, b_2 \in \{1, \ldots, N\}$ such that $U^{b_1 b_2}_{a_1 a_2} = 1$, and all other matrix elements are zero. Thus the classical unitary operator corresponds to a bijective mapping $X^2 \to X^2$ with $X = \{1, \ldots, N\}$, defined by the functions $b_1(a_1, a_2)$ and $b_2(a_1, a_2)$. The operator corresponds to the quantum state given by

$$|\psi\rangle = \frac{1}{N} \sum_{a_1, a_2 = 1}^{N} |a_1, a_2; b_1, b_2\rangle, \qquad b_1 = b_1(a_1, a_2), \ b_2 = b_2(a_1, a_2). \tag{73}$$

Such states have *minimal support*: It is not possible to satisfy the unitarity condition with states with a fewer number of non-zero components.

The components of the state (73) define classical configurations of the square, where each vertex is assigned a concrete value. We call the quartet of values $(a_1, a_2, b_1, b_2)$ an allowed configuration in the state (73) if the corresponding basis element $|a_1, a_2; b_1, b_2\rangle$ appears in (73).

Now the dual unitarity condition is the following: for each pair $(a_1, b_2)$ there is exactly one pair $(a_2, b_1)$ such that the quartet $(a_1, a_2, b_1, b_2)$ is an allowed configuration, and the same holds also vice versa. In other words, the set of quartets defines a bijection between the pairs $(a_1, b_2)$ and $(a_2, b_1)$. Combining the unitarity requirements we obtain the following criterion: A deterministic operator is dual unitary if the set of the allowed configurations is such that a quartet $(a_1, a_2, b_1, b_2)$ can be uniquely identified by the pairs $(a_1, a_2)$, or $(a_2, b_1)$, or $(b_1, b_2)$ or $(a_1, b_2)$. The choice of these pairs comes from all the possibilities of selecting two neighbouring sites of the square.

Let us also consider the hexagonal geometry. Now the unitary matrix given by components $U^{b_1 b_2 b_3}_{a_1 a_2 a_3}$ is such that for each triplet of "input variables" $(a_1, a_2, a_3)$ there is a deterministic triplet of "output variables" $(b_1, b_2, b_3)$ such that each $b_j$ is a function $b_j(a_1, a_2, a_3)$. Furthermore, the mapping between the triplets is bijective. In such a case we construct the vector

$$|\psi\rangle = \frac{1}{N^{3/2}} \sum_{a_1, a_2, a_3 = 1}^{N} |a_1, a_2, a_3; b_1, b_2, b_3\rangle. \tag{74}$$

Now we call a sextet $(a_1, a_2, a_3, b_1, b_2, b_3)$ an allowed configuration in the state (74) if the corresponding basis element $|a_1, a_2, a_3; b_1, b_2, b_2\rangle$ appears in (74). We associate the variables with the six vertices of a hexagon, such that they are written down in an anti-clockwise manner (similar to the labels in Fig 2).

Hexagonal unitarity requires that rotations of the state (74) also correspond to unitary operators. From this we deduce a condition for the sextets: A state (74) is hexagonal unitary if at any three neighbouring sites of the hexagon any of the $N^3$ possible triplets of values appears in exactly one allowed configuration.

For such states it is also customary to compile a table out of the allowed configurations. For example, in the case of the dual unitary matrices the resulting table is of size $N^2 \times 4$ and it includes the quartets $(a_1, a_2, b_1, b_2)$. In the hexagonal cases the table includes the sextets $(a_1, a_2, a_3, b_1, b_2, b_3)$ and it becomes of size $N^3 \times 6$. More generally, if there are a total number of $K$ variables, then we obtain a table of size $N^{K/2} \times K$. These tables are generalizations of the *orthogonal arrays* known from combinatorial design theory. In the simplest case an orthogonal array of strength $t$ with $N$ levels and $k$ factors is a table with $N^t$ rows, $k$ columns, filled with numbers $1, \ldots, N$ with the following "orthogonality" property: if one selects $t$ columns of the table in *any possible way* then in the resulting sub-table with $t$ columns and $N^t$ rows, every one of the $N^t$ $t$-tuplets is present in exactly one row.

The tables constructed from the multi-directional unitary matrices satisfy a relaxed version of the above "orthogonality" property: if one selects $t$ columns of the table *in certain allowed ways*, then in the resulting sub-table with $t$ columns and $N^t$ rows, every one of the $N^t$ $t$-tuplets is present in exactly one row. The allowed subsets of columns coincide with the allowed subsystems in (16). In the case of the planar arrangements these tables were already discussed in [45], where they were called "planar orthogonal arrays". Our concept here is even more general because we also allow higher dimensional arrangements, such as the cubic arrangement in three space dimensions.

In the most general case our state is of the form

$$|\psi\rangle = \frac{1}{N^{K/2}} \sum_{a_1, \ldots, a_{K/2} = 1}^{N} \left| a_1, \ldots, a_{K/2}; b_1, \ldots, b_{K/2} \right\rangle, \tag{75}$$

where it is understood that each $b_j$ is a function of the tuple $(a_1, \ldots, a_{K/2})$. In such expressions the ordering of the variables is such that $a_j$ and $b_j$ are sitting on antipodal sites on the same diagonal, and the ordering of the diagonals is the same as laid out in Section 2.5.

## 7.1 Spatially symmetric solutions

Let us turn again to the spatially symmetric cases. Classical solutions with spatial symmetries can be found by studying the orbits of classical configurations under the geometric symmetry group. Let us denote by $|\phi\rangle$ a classical configuration, that is

$$|\phi\rangle = |c_1, c_2, \ldots, c_K\rangle, \qquad c_j \in \{1, \ldots, N\}. \tag{76}$$

We define its orbit as

$$o(|\phi\rangle) = \sum_k |\phi_k\rangle, \tag{77}$$

where the summation is over all such (distinct) states $|\phi_k\rangle$, which can be obtained from $|\phi\rangle$ using a symmetry transformation. More precisely, the summation includes states $|\phi_k\rangle$ for which there is a $g \in G$ such that $\mathcal{P}_g |\phi\rangle = |\phi_k\rangle$. It is important that the summation is over the configurations, and not the symmetry transformations. This way each vector $|\phi_k\rangle$ is present in the summation only one time, with a coefficient of 1.

Let us consider examples for orbits. We take $N = 2$ and the geometry of the square. Four examples for orbits are given by

$$
\begin{aligned}
o(|1111\rangle) &= |1111\rangle, \\
o(|1212\rangle) &= |1212\rangle + |2121\rangle, \\
o(|1122\rangle) &= |1122\rangle + |1221\rangle + |2211\rangle + |2112\rangle, \\
o(|1112\rangle) &= |1112\rangle + |1121\rangle + |1211\rangle + |2111\rangle.
\end{aligned}
\tag{78}
$$

We encourage the reader to check that these are indeed the orbits, by using the ordering of the sites laid out in Section 2.5.

Clearly, spatially symmetric states $|\psi\rangle$ can be constructed as linear combinations of different orbits. Let us now take a set of configurations $\{|\phi_k\rangle\}_{k=1,\ldots,n}$ such that all of them belong to different orbits, and construct the desired state as

$$|\psi\rangle = \frac{1}{N^{K/2}} \sum_{k=1}^{n_o} o(|\phi_k\rangle). \tag{79}$$

Here we did not specify the number $n_o$ of the different orbits that are needed (this will depend on the state), and we included a normalization factor in anticipation of the final result. A state defined as (79) is manifestly symmetric, but it is not yet established that it describes a multi-directional unitary operator.

## 7.2 Multi-directional unitarity

Now we explore the unitarity conditions for the state (79). It follows from the spatial symmetry that unitarity needs to be checked with respect to only one of the allowed bipartitions. The unitarity condition will be satisfied automatically for the other bipartitions.

Returning to the state (75), we see that it describes a unitary matrix if it describes a bijection between the tuplets $(a_1, \ldots, a_{K/2})$ and $(b_1, \ldots, b_{K/2})$. This means that for every tuplet $(a_1, \ldots, a_{K/2})$ there is precisely one allowed configuration in the state which has the numbers $(a_1, \ldots, a_{K/2})$ as the first $K/2$ numbers, and also that for each tuplet $(b_1, \ldots, b_{K/2})$ there is precisely one allowed configuration which has these numbers as the last $K/2$ numbers. This

condition is easy to check once an explicit form for (75) is given. However, it is less evident from the expression (79).

In order to find and describe the solutions we need to introduce new definitions.

We say that an orbit (77) is non-overlapping if there are no two distinct configurations $|\phi_1\rangle$ and $|\phi_2\rangle$ which both appear in the orbit and which have the same values for the first $K/2$ variables. It is clear that a state $|\psi\rangle$ with the desired entanglement properties can include only non-overlapping orbits, because otherwise the classical unitarity would be broken already by this single orbit.

For example, consider the geometry of the square, $N = 2$, and the orbits given in eq. (78). It can be checked that the first three orbits are non-overlapping, but the fourth orbit is overlapping. In that case the first two configurations have the same pair $(1, 1)$ as the first two components. Therefore, this fourth orbit cannot be used for our purposes.

We say that two orbits are mutually non-overlapping if there are no two configurations $|\phi_1\rangle$ and $|\phi_2\rangle$ from the first and the second orbit, respectively, such that $|\phi_1\rangle$ and $|\phi_2\rangle$ have the same values for the first $K/2$ components. States $|\psi\rangle$ with the desired properties can be obtained as sums of orbits which are non-overlapping and also mutually non-overlapping.

The last property that needs to be checked is that for each possible value of the tuplet $(a_1, a_2, \ldots, a_{K/2})$ there is an orbit containing an allowed configuration which includes these numbers as the first $K/2$ variables. We call this the criterion of completeness.

Having clarified these conditions it is straightforward to set up an algorithm to find the solutions. For each geometry and each $N$ one needs to determine the list of orbits which are non-overlapping. Having determined this list one needs to look for a selection of orbits such that each two selected orbits are mutually non-overlapping, and the criterion of completeness is also satisfied. In simple cases these computations can be carried out by hand. In fact we found certain solutions without the help of a computer. On the other hand, for bigger values of $K$ and $N$ it is useful to write computer programs, so that all solutions can be found.

Let us discuss a particular solution as an example. We take the square geometry and $N = 4$. We start with the orbit

$$o(|1234\rangle) = |1234\rangle + |2341\rangle + |3412\rangle + |4123\rangle + |4321\rangle + |3214\rangle + |2143\rangle + |1432\rangle. \quad (80)$$

It can be seen that this orbit is non-overlapping. We select the following also non-overlapping orbits:

$$
\begin{aligned}
o(|aaaa\rangle) &= |aaaa\rangle, \qquad a = 1, 2, 3, 4, \\
o(|1313\rangle) &= |1313\rangle + |3131\rangle, \\
o(|2424\rangle) &= |2424\rangle + |4242\rangle.
\end{aligned}
\quad (81)
$$

Once again, all six orbits are non-overlapping. Furthermore, we can observe that all of the seven orbits are mutually non-overlapping. Finally, their sum is complete. This means that the state $|\psi\rangle$ given by

$$
\begin{aligned}
|\psi\rangle = \frac{1}{4}\Big[ &|1234\rangle + |2341\rangle + |3412\rangle + |4123\rangle \\
&+ |4321\rangle + |3214\rangle + |2143\rangle + |1432\rangle \\
&+ |1313\rangle + |3131\rangle + |2424\rangle + |4242\rangle \\
&+ |1111\rangle + |2222\rangle + |3333\rangle + |4444\rangle \Big],
\end{aligned}
\quad (82)
$$

satisfies both the symmetry and the unitarity requirements.

The state above is specified by its seven orbits. However, six of these orbits consist of configurations which correspond to identity operations along the diagonals. In order to simplify

our presentation we introduce a separate name for such cases. We say that a configuration is diagonally identical if it is of the form

$$\left|a_1,\ldots,a_{K/2};b_1,\ldots,b_{K/2}\right\rangle = \left|a_1,\ldots,a_{K/2};a_1,\ldots,a_{K/2}\right\rangle. \tag{83}$$

This means that two variables on the same diagonal always have the same value. The orbit of a diagonally identical configuration consists of configurations of the same type. Such orbits are always non-overlapping. What is more, two different orbits of the same type are always mutually non-overlapping.

Most of the solutions that we find will have a number of diagonally identical orbits. This was demonstrated in the example (82), which has 6 diagonally identical orbits, and only 1 orbit which is of different type.

When describing the solutions we will always present the list of orbits which are not diagonally identical. Once such a list is given, it is straightforward to complement the state with diagonally identical orbits. The algorithm for this is the following: We start with the sum over the orbits which are not diagonally identical. For each tuplet $(a_1,\ldots,a_{K/2})$ with $a_j = 1,\ldots,N$, $j = 1,\ldots,K/2$ we check whether there is an allowed configuration in the sum of orbits which has these values as the first $K/2$ variables. If there is no such configuration, then we add the diagonally identical orbit of $(a_1,\ldots,a_{K/2})$ to the state. Afterwards we proceed with checking the tuplets, and eventually this leads to a complete solution.

Let us demonstrate this procedure on the example of the state (82). Now the only orbit which is not diagonally identical comes from the configuration $|1234\rangle$. This orbit gives the first 8 terms in (82). Now we perform the check over the pair of variables $(a_1, a_2)$. There are 16 such pairs, and we see that 8 of them are present in the first orbit. The remaining ones are $(1,3), (3,1), (2,4), (4,2)$ and the four pairs $(a,a)$ with $a = 1,\ldots,4$. The resulting diagonally identical orbits are

$$
\begin{aligned}
o(|1313\rangle) &= |1313\rangle + |3131\rangle, \\
o(|2424\rangle) &= |2424\rangle + |4242\rangle,
\end{aligned} \tag{84}
$$

and the four orbits with the single element $|aaaa\rangle$ with $a = 1,\ldots,4$. Adding these orbits we obtain the state (82).

This demonstrates that indeed it is enough to list the diagonally not-identical orbits. In order to further simplify the notation, we will label an orbit as $[a_1a_2\ldots b_1b_2\ldots]$, and we will simply just list such labels. In this notation the state (82) is "encoded" simply as $[1234]$. This means that the state is constructed from the orbit $o(|1234\rangle)$ and the completion as described above. For clarity we also treat another example. Again for $K = 4$ and $N = 4$ consider the list of orbits

$$[1424], [3344]. \tag{85}$$

These two labels encode a solution as follows. The orbits are given by

$$
\begin{aligned}
o(|1424\rangle) &= |1424\rangle + |2414\rangle + |4142\rangle + |4241\rangle, \\
o(|3344\rangle) &= |3344\rangle + |3443\rangle + |4433\rangle + |4334\rangle.
\end{aligned} \tag{86}
$$

We complete these configurations as

$$
\begin{aligned}
|\psi\rangle = \frac{1}{4}\Big[ &|1424\rangle + |2414\rangle + |4142\rangle + |4241\rangle \\
&+ |3344\rangle + |3443\rangle + |4433\rangle + |4334\rangle \\
&+ |1212\rangle + |2121\rangle + |1313\rangle + |3131\rangle \\
&+ |2323\rangle + |3232\rangle + |1111\rangle + |2222\rangle \Big].
\end{aligned} \tag{87}
$$

Now the first two lines come from the diagonally not-identical orbits given in the list (85), and the last two lines come from the completion. Therefore, the short notation (85) completely describes the final state (87), which is a solution to all our requirements.

Other examples for the expansion of the list of orbits into the full state are presented in Appendix B.

We performed a search for all solutions using computer programs. We considered all geometries and small values of $N$. In each case we increased $N$ until we found non-trivial solutions, which are not related to the identity matrix. All results that we report below are exhaustive and have been crosschecked using at least two independent algorithms. The first method was to generate a list of all non-overlapping orbits, and then search for combinations of mutually non-overlapping orbits from this list. The second method was a simple adaptation of the algorithm used in [60] for generating equivalence classes of orthogonal arrays.

Before turning to concrete solutions we also discuss the local equivalences between the solutions.

### 7.3 Equivalence classes

It was stated in Section 2.7 that a natural LU equivalence between two spatially symmetric states is the one given by (28). However, in the classical case it is more natural to allow only classical equivalence steps. Therefore, we say that two classical solutions $|\psi_1\rangle$ and $|\psi_2\rangle$ are "weakly equivalent" if there exists a one-site permutation operator $V$, such that

$$|\psi\rangle \quad \rightarrow \quad (V \otimes V \otimes \cdots \otimes V)|\psi\rangle. \tag{88}$$

Such a symmetric transformation preserves the structure of the orbits, it just re-labels the variables.

Alternatively, we can also require stronger equivalence relations. One option is to allow the connection (28) using arbitrary $SU(N)$ operators (not just permutations), but still requiring exact spatial symmetry. The other option is to allow permutation operators, but to drop the spatial symmetry and allow a generic situation as in (26). This gives

$$|\psi\rangle \quad \rightarrow \quad (A^{(1)} \otimes A^{(2)} \otimes \cdots \otimes A^{(K)})|\psi\rangle, \tag{89}$$

where now each $A^{(j)}$ is a one-site permutation matrix. These stronger equivalence relations can connect classical states which appear different if we consider the weak equivalence. On the other hand, they can also change the structure of the orbits.

Consider for example the following dual unitary operator for $N = 2$:

$$U = X \otimes X, \tag{90}$$

where $X$ denotes the respective Pauli matrix. The operator (90) describes a spin-flip along the diagonals of the square. Therefore it is trivial: there is no entanglement between the diagonals. Furthermore, it is classical: it is represented by the state

$$|\psi_1\rangle = \frac{1}{2}o(|1122\rangle) = \frac{1}{2}\Big[|1122\rangle + |1221\rangle + |2112\rangle + |2211\rangle\Big]. \tag{91}$$

On the other hand, we can also consider the diagonally identical state corresponding to $U = 1$:

$$|\psi_2\rangle = \frac{1}{2}\Big[|1111\rangle + |1212\rangle + |2121\rangle + |2222\rangle\Big]. \tag{92}$$

The two states are not LU equivalent with respect to (88): for $N = 2$ the only one-site permutations are the identity and the spin-flip, and their symmetric application leaves both states

invariant. On the other hand, the strong equivalence steps do connect them. Regarding the symmetric but quantum mechanical equivalence we can use the operator

$$V = \frac{1-i}{2}\begin{pmatrix} 1 & i \\ i & 1 \end{pmatrix}, \qquad V^2 = X. \tag{93}$$

Taking the "square root" of (90) we see that the equivalence (28) holds with this choice for $V$. Alternatively, we can also connect $|\psi_1\rangle$ and $|\psi_2\rangle$ using a non-symmetric permutation equivalence, for example

$$|\psi_1\rangle = (X \otimes X \otimes 1 \otimes 1)|\psi_2\rangle. \tag{94}$$

In this example the structure of the orbits is also changed. The state $|\psi_1\rangle$ is a single orbit, whereas $|\psi_2\rangle$ consists of three diagonally identical orbits.

We define the weak and strong equivalence classes according to the equivalence steps discussed above. The two equivalence classes are generally different, the strong equivalence classes can split into different weak classes.

Below we list the solutions up to the strong equivalence. This means that for each geometry and each $N$ we give a list of states such that no two states can be transformed into each other via (89).

For each equivalence class we choose a representative such that it has the fewest orbits that are not diagonally identical within that class. Returning to the examples of the states $|\psi_1\rangle$ and $|\psi_2\rangle$ we would choose $|\psi_2\rangle$ because this state has only diagonally identical orbits. In other words, we choose representatives which are closest to the identity operation.

## 7.4 Dual unitary matrices

For $N = 2$ we found that there is only one equivalence class, namely the one given by the identity operator. Note that using strong equivalence this class includes solutions such as (90).

For $N = 3$ there are two equivalence classes, namely the one given by the identity operator, and another one given by [1122] (which, unlike (90) in the case of $N = 2$, is not equivalent to the identity).

A range of new solutions appear at $N = 4$. These solutions are:

1. Identity;

2. [1234];

3. [1122];

4. [1424];

5. [1424], [2343];

6. [1424], [2343], [1122];

7. [1424], [2343], [1213];

8. [1424], [2233];

9. [1424], [3344];

10. [1424], [3344], [1232].

Apparently these solutions arise as sums of different orbits taken from a small list of orbits. Again, they each define an equivalence class with respect to the transformations (89).

It is straightforward to use our computer programs to search for matrices with higher $N$, and thereby perform a partial enumeration. However, the runtime of the program and the resulting list of solutions quickly becomes too long. This is an example of a phenomenon known as the "combinatorial explosion". It makes it difficult to even get meaningful estimates of the number of solutions for higher $N$. Therefore we only publish results for small $N$, for which the list of equivalence classes is short enough to include in the article.

## 7.5 Hexagonal geometry

For $N = 2$ there is only a single equivalence class, which is the class of the identity matrix.

New solutions appear at $N = 3$. The full list of the solutions (strong equivalence classes) for $N = 3$ is

1. Identity;

2. [122 322];

3. [122 322], [133 233];

4. [122 322], [133 233], [211 311];

5. [222 333], [232 323];

6. [122 133], [123 132];

7. [122 133], [232 323], [212 333], [113 223], [131 313];

8. [111 232], [222 313], [333 121];

9. [122 213], [232 323], [123 332], [132, 231];

10. [111 222], [121 323], [212 333].

As an example, Appendix B shows how to obtain the full state from the list of orbits in the case of solution No. 10.

## 7.6 Cubic geometry

The case of the cube is special: there are non-trivial solutions even in the case of $N = 2$. There are two strong equivalence classes, each having two solutions. One contains the identity and [1112 2221], [1122 2211], [1212 2121]. The other contains [1112 2221] and [1122 2211], [1212 2121].

As an example, Appendix B shows how to obtain the full state for [1112 2221].

## 7.7 Octahedral geometry

For $N = 2$ there is just one strong equivalence class, namely the one given by the identity operator.

New solutions appear for $N = 3$. The list of solutions (strong equivalence classes) is

1. Identity;

2. [222 333];

3. [331 221];

4. [331 332];

5. [331 332], [221 223];

6. [331 332], [221 223], [112 113].

As an example, Appendix B shows how to obtain the full state from the list of orbits in the case of solution No. 6.

Note that each one of these cases also appears in the list of the hexagonal solutions. There are only 6 octahedral solutions instead of the 10 hexagonal solutions: they are those cases which also satisfy the extra unitarity condition on top of those of the hexagonal case. The list number of the matching hexagonal solutions is the following: 1: 1, 2: 5, 3: 6, 4: 2, 5: 3, 6: 4. (The list of orbits is different in some solutions because octahedral orbits are different from hexagonal ones.)

## 8   Summary and outlook

In this work we gave a unified treatment of multi-directional unitary operators and their corresponding multi-directional maximally entangled states. Using this unified treatment, we have presented various ways to design such operators. Most of these constructions had already appeared in the literature before, but our treatment is unique: We centered the discussion around the geometric properties of the states/operators, and this way we could treat all the different constructions within the same formalism. Moreover, we believe that one of the geometries, namely the octahedral geometry has not been considered before.

We focused on spatially symmetric solutions and in many cases we have found new concrete examples of multi-directional unitary operators. Our new results include

- The observation that the octahedral case is a specialization of the hexagonal case (with one more unitarity constraint).

- Pointing out that there exists an AME of six qubits with exact hexagonal symmetry.

- Finding graph states for the hexagonal, cubic and octahedral geometries.

- Extending the construction of [53] with Hadamard matrices to the cubic case.

- Determining the classical solutions for small values of $N$ in all geometries.

There are various open problems which are worth pursuing.

First of all, it would be interesting to find a complete description of the algebraic varieties at least in some simple cases, perhaps with additional symmetry properties. So far the only complete description is for the four qubits in the square geometry (dual unitary matrices) found in [3]. It is likely that complete solutions (explicit parametrizations) will not be found in other cases unless further restrictions are imposed. At present it is not clear how to extend the methods of [3] to the higher dimensional or multi-leg cases.

It is also interesting to consider spatial invariance in a weak sense, as described at the end of Subsection 2.7 and in the Appendix A. Allowing a combination of permutations and LU operations opens up more possibilities for such states.

Another interesting question is factorization in the cases with $K \geqslant 6$. Earlier works already noted that multi-directional unitary operators can be constructed from products of the same objects with fewer sites (tensor legs). For example, the work [36] included a "braiding" construction for states in the geometry of a perfect polygon with $K \geqslant 6$ sites, such that each braiding step involves a dual unitary operator (this idea was originally suggested by Vaughan

Jones). Similarly, gates for the cubic geometry were constructed in [19] with a product of a small number of dual unitary operators. Similar factorizations can be worked out in many cases, with small variations. Then it is also an important question: Which final solutions can be factorized? In this work we did not touch this question, except for pointing out that the operator in formula (43) can be factorized in a special case. It is likely that most of our solutions cannot be factorized, nevertheless this question deserves a proper study. Finding conditions for factorizability and/or actually proving the impossibility of factorization in certain cases would help the general understanding of these objects. Several of these questions are considered in a follow-up paper [61].

In this work we focused only on the multi-directional unitary operators, and not on their applications. It would be interesting to study the quantum circuits that arise from these objects. With the exception of the dual unitary circuits, there has not been much progress yet. General statements about the behaviour of two-point functions were already given in [18,19], but the dynamics of entanglement generation has not yet been investigated. It would be interesting if certain models could be identified, where the entanglement evolution could be computed, similar to some special dual unitary models such as the kicked Ising model.

## Acknowledgments

We are thankful for discussions with János Asbóth, Wojciech Bruzda, Máté Matolcsi, Milán Mosonyi, Péter Vrana, Mihály Weiner, Tianci Zhou, Zoltán Zimborás, and Karol Życzkowski.

## A    Spatial invariance in a weak sense

Here we provide two examples for a phenomenon discussed in Section 2.7: states can have spatial invariance in a weak sense, such that the spatial symmetry operations transform them into other vectors that are LU equivalent to themselves.

The simplest example is in the case of the square geometry, which corresponds to the dual unitary operators. Now we present the example on the level of the states. We consider four qubits, and two $SU(2)$-singlets which are prepared on pairs of opposite sites, i.e. there is one singlet for each diagonal of the square. This gives (with some abuse of notation)

$$|\psi\rangle = \left(\frac{|12\rangle - |21\rangle}{\sqrt{2}}\right)_{13}\left(\frac{|12\rangle - |21\rangle}{\sqrt{2}}\right)_{24} = \frac{1}{2}\left(|1122\rangle - |1221\rangle - |2112\rangle + |2211\rangle\right). \quad (A.1)$$

This state has maximal entanglement for the two bipartitions dictated by the geometry (namely $\{1,2\} \cup \{3,4\}$ and $\{1,4\} \cup \{2,3\}$). Furthermore, it is invariant with respect to reflections of the square. However, a rotation of the square by $\pi/2$ results in $-|\psi\rangle$, thus we obtain a "projective representation" of the invariance property.

Another example is given by a well known AME with $N = 3$ and four parties. It is given by

$$|\psi\rangle = \sum_{a,b=0}^{2} |a, b, a + b, a - b\rangle. \quad (A.2)$$

Here the algebraic operations are performed in the finite field $\mathbb{Z}_3$, and the indices for the basis vectors run from 0 to 2. This state has maximal entanglement for all bipartitions. And it is symmetric in a weak sense: any permutation of its sites can be "undone" by local unitary transformations. For example, consider the permutation of the first two sites. After a change

of indices $a \longleftrightarrow b$ we obtain the new representation

$$\mathcal{P}_{12}|\psi\rangle = \sum_{a,b=0}^{2} |a, b, a + b, -(a - b)\rangle. \tag{A.3}$$

We see that the only difference is in the last component. However, this can be transformed away by a unitary operator acting on the last site, which performs the change of basis elements $|1\rangle \longleftrightarrow |2\rangle$, corresponding to negation in $\mathbb{Z}_3$. It can be checked that similar one-site transformations (or combinations thereof) exist for all rearrangements of the sites. Therefore the state (A.2) is invariant in a weak sense.

# B   Explicit formulas for classical solutions

In Subsections 7.4–7.7 we give lists of classical multi-directional unitary operators arranged in various geometries. These lists contain all the classical multi-directional unitary operators with certain low qudit dimensions $N$ that are completely symmetric with respect to their geometric arrangement. The operators in these lists are written in a compact notation which enumerates only the diagonally non-identical orbits (for an explanation see the paragraphs following (82)). In this Appendix we show examples of the expansion of this compact notation into the quantum state describing the operator. In the case of the square geometry (or dual unitary operators) an example is given in (85)–(87). In this Appendix we consider the other three geometries.

**Hexagonal geometry**   Let us consider solution No. 10. in the list in Subsection 7.5 with non-diagonal orbits [111 222], [121 323], [212 333]. The corresponding state is

$$
\begin{aligned}
|\psi\rangle = \frac{1}{3\sqrt{3}}\Big[ &\big(|111\,222\rangle + |112\,221\rangle + |122\,211\rangle + |222\,111\rangle + |221\,112\rangle + |211\,122\rangle\big) \\
&+ \big(|121\,323\rangle + |213\,231\rangle + |132\,312\rangle + |323\,121\rangle + |231\,213\rangle + |312\,132\rangle\big) \\
&+ \big(|212\,333\rangle + |123\,332\rangle + |233\,321\rangle + |333\,212\rangle + |332\,123\rangle + |321\,233\rangle\big) \\
&+ \big(|113\,113\rangle + |131\,131\rangle + |311\,311\rangle\big) + \big(|313\,313\rangle + |133\,133\rangle + |331\,331\rangle\big) \\
&+ \big(|232\,232\rangle + |322\,322\rangle + |223\,223\rangle\big)\Big].
\end{aligned}
\tag{B.1}
$$

In the r.h.s. distinct orbits are separated by parentheses. The first three lines of the r.h.s. contain the three diagonally non-identical orbits. The last two lines contain diagonally identical orbits.

**Cubic geometry**   Let us consider one of the non-trivial solutions that appear in Subsection 7.6 for $N = 2$: [1112 2221]. The corresponding state is

$$
\begin{aligned}
|\psi\rangle = \frac{1}{4}\Big[ &\big(|1112\,2221\rangle + |1121\,2212\rangle + |1211\,2122\rangle + |2111\,1222\rangle \\
&+ |2221\,1112\rangle + |2212\,1121\rangle + |2122\,1211\rangle + |1222\,2111\rangle\big) \\
&+ \big(|1111\,1111\rangle\big) + \big(|2222\,2222\rangle\big) + \big(|1212\,1212\rangle + |2121\,2121\rangle\big) \\
&+ \big(|1122\,1122\rangle + |1221\,1221\rangle + |2211\,2211\rangle + |2112\,2112\rangle\big)\Big].
\end{aligned}
\tag{B.2}
$$

Again, distinct orbits are separated by parentheses. The first two lines contain the diagonally non-identical orbit. The rest contains identical orbits.

**Octahedral geometry**  Let us consider solution No. 6. in the list in Subsection 7.7. The diagonally non-identical orbits are [331 332], [221 223], [112 113].

$$
\begin{aligned}
|\psi\rangle = \frac{1}{3\sqrt{3}}\Big[ &\big(|331\,332\rangle + |332\,331\rangle + |313\,323\rangle + |323\,313\rangle + |133\,233\rangle + |233\,133\rangle\big) \\
&+ \big(|221\,223\rangle + |223\,221\rangle + |212\,232\rangle + |232\,212\rangle + |122\,322\rangle + |322\,122\rangle\big) \\
&+ \big(|112\,113\rangle + |113\,112\rangle + |121\,131\rangle + |131\,121\rangle + |211\,311\rangle + |311\,211\rangle\big) \\
&+ \big(|111\,111\rangle\big) + \big(|222\,222\rangle\big) + \big(|333\,333\rangle\big) \\
&+ \big(|123\,123\rangle + |231\,231\rangle + |312\,312\rangle + |321\,321\rangle + |213\,213\rangle + |132\,132\rangle\big) \Big].
\end{aligned}
$$

(B.3)

Distinct orbits are separated by parentheses. The first three lines contain the three diagonally non-identical orbits, the rest contains identity orbits.

Since the hexagonal constraints form a subset of the octahedral constraints, the state (B.3) also corresponds to the hexagonal unitary operator No. 4 in Subsection 7.5. Under the weaker hexagonal symmetry, the orbit in the last line, [123 123], splits into two orbits: [123 123] and [132 132].

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
