# Peer review of "Multi-directional unitarity and maximal entanglement in spatially symmetric quantum states"

_SciPost Physics, doi:SciPost Phys. 16, 010 (2024)_

## Round 1 · Referee Report · Anonymous (Referee 1) · 2023-11-9

Report

The manuscript "Multi-directional unitary and maximal entanglement in spatially symmetric quantum states" generalizes the notion of dual unitary matrices to unitaries with similar properties in different geometries, dubbed "multi-directional unitaries".
After recalling the definition of dual unitaries (which are unitaries acting on bipartite systems and keep being unitary under a certain rearrangement of the matrix elements), and their connection to highly entangled states using the operator-state correspondence, the concept of multi-directional unitaries is introduced via a direct generalization. This generalization requires an underlying geometry which defines the rearrangements under which the matrix shall keep being unitary. The considered geometries are squares, hexagons, regular polygons, cubes, octahedra and tetrahedra, as the corresponding multi-directional unitaries have had use cases in the literature.
The main body of the manuscript is then considered with different constructions for thee multi-directional unitaries, starting with trivial ones (the corresponding states of which are tensor products of Bell states), followed by constructions using diagonal unitary matrices, Hadamard matrices, graph states and "classical" constructions, involving unitaries which are represented by permutation matrices.

The manuscript is very well written, in particular, the review of existing literature on related topics, such as AME states and dual unitaries, is extensive. The derivations are correct and the topic is of multi-directional unitaries is certainly of relevance, as they are used, for example, in the construction of quantum cellular automata and appear in the kicked Ising model. As such, the plethora of constructions introduced will be certainly useful in the future, There are, however, some minor points of critique which should be addressed by the authors, which are given below. After an appropriate revision, I recommend publication of the manuscript in SciPost.

Requested changes

  1. Using both, Ǔ and U for related matrices tends to be confusing (see also below), especially given the fact that the used operator-state correspondence given in Eq. 2.7 differs from the "standard" Choi correspondence, which is given in Eq. 2.17. Is the introduction of Ǔ really necessary? It seems that the results can also be formulated in terms of U instead of Ǔ, or not? If this is not the case, please justify the use of both notations more than just writing "On the other hand, we also introduce the operator U via...".

  2. In Eq. 3.1 and 3.2, I think that PUP should read PǓP instead: P...P is a horizontal along the center, which certainly is part of the square symmetry.

  3. After Eq. 5.7, "we see that U is indeed unitary" should read "we see that Ǔ is indeed unitary".

  4. In Cpt. 6.1, I do not see why the solution alpha = 1 and beta = 2 requires N to be prime if N > 3. Even in non-prime dimensions, the condition beta^2 not equal to alpha^2 is fulfilled.

---

## Round 1 · Referee Report · Anonymous (Referee 2) · 2023-11-17

Report

The text discusses the concept of "multi-directional unitarity" and "multi-directional maximally entangled states" in quantum mechanics. These objects are generalizations of dual unitary operators and have special entanglement properties relevant to quantum many-body physics and quantum information theory. The focus is on quantum states with maximal entanglement for all bipartitions that follow from the reflection symmetries of a spatially symmetric arrangement of sites.
The study extends the notion of "dual unitarity" in solvable quantum many-body systems, introducing the concept of multi-directional unitarity in various geometries, including hexagonal, cubic, and octahedral. The text relates these concepts to the broader field of quantum information theory, particularly the study of "absolutely maximally entangled states" (AME), which have maximal bipartite entanglement for all possible bipartitions. AMEs are explored for their applications in quantum error correction codes and tensor network models.
The authors present a unified treatment of multi-directional unitary operators, emphasizing geometric properties and providing various constructions and examples. The discussion includes octahedral geometry, which has not been considered before and highlights spatially symmetric solutions. The text concludes with open problems, suggesting avenues for further research, such as finding complete descriptions of algebraic varieties, considering spatial invariance, exploring factorization in cases with more sites, and studying the quantum circuits arising from these operators.

Just some final comments, we found some small typos that can be corrected during the publication of the article.

All in all, we recommend the publication of the paper.

---

## Round 2 · Referee Report · Anonymous · 2023-12-8

Report

Thank you for considering my previous comments. In principal, I am happy with the updated manuscript and proposed changes. However, in this updated version 2, I can see only the added footnote, but the typos in Eqs. 3.1 and 3.2 and after Eq. 5.7 seem to be still present. Please correct them properly. As soon as this is done, the manuscript is ready for publication from my point of view.

---

## Round 2 · Referee Report · Anonymous · 2023-12-13

Report

After reading the second version of the manuscript and the reply to the referees, we recommend the publication as an article

---

## Round 2 · Author Response

We are thankful to the referees for the review and the comments.

---

## Round 2 · List of Changes

Referee 1 had 4 requests. Here are our replies:
1. We added a footnote about the reason, why we introduce both $U$ and $\check U$, at the position where $U$ is introduced.
2. Indeed, these were small typos. Now corrected.
3. Again, a small typo, now corrected.
4. The general derivations of graphs states depend on $N$ being a prime number. We write this in the first paragraph of the Section. We did not want to discuss all the derivations, to make clear what holds and what does not hold if $N$ is prime. So all of the Section focuses on $N$ being prime. We are not sure how to make this more pronounced, so we left this as it is.

---

## Round 3 · Referee Report · Anonymous (Referee 5) · 2023-12-14

Report

Thank you for correcting the mistakes. This version is ready for publication.

---

## Round 3 · Author Response

We are thankful the referees for the renewed review, and especially to referee 1. for checking every detail again. We are honestly sorry for the extra time/work needed. It is not clear what happened, different versions of the tex file got mixed up somehow. In any case, the current version should be fine.

---

## Round 3 · List of Changes

Eqs. (3.1), (3.2) and below (5.7) are fine now.

---

## Editorial Decision

published